

# Arrival and diversification of mabuyine skinks (Squamata: Scincidae) in the Neotropics based on a fossil-calibrated timetree

Anieli Guirro Pereira and Carlos G. Schrago

Department of Genetics, Federal University of Rio de Janeiro, Rio de Janeiro, RJ, Brazil

## ABSTRACT

**Background**. The evolution of South American Mabuyinae skinks holds significant biogeographic interest because its sister lineage is distributed across the African continent and adjacent islands. Moreover, at least one insular species, *Trachylepis atlantica*, has independently reached the New World through transoceanic dispersal. To clarify the evolutionary history of both Neotropical lineages, this study aimed to infer an updated timescale using the largest species and gene sampling dataset ever assembled for this group. By extending the analysis to the Scincidae family, we could employ fossil information to estimate mabuyinae divergence times and carried out a formal statistical biogeography analysis. To unveil macroevolutionary patterns, we also inferred diversification rates for this lineage and evaluated whether the colonization of South American continent significantly altered the mode of Mabuyinae evolution.

**Methods**. A time-calibrated phylogeny was inferred under the Bayesian framework employing fossil information. This timetree was used to (i) evaluate the historical biogeography of mabuiyines using the statistical approach implemented in Bio-GeoBEARS; (ii) estimate macroevolutionary diversification rates of the South American Mabuyinae lineages and the patterns of evolution of selected traits, namely, the mode of reproduction, body mass and snout–vent length; (iii) test the hypothesis of differential macroevolutionary patterns in South American lineages in BAMM and GeoSSE; and (iv) re-evaluate the ancestral state of the mode of reproduction of mabuyines.

**Results**. Our results corroborated the hypothesis that the occupation of the South American continent by Mabuyinae consisted of two independent dispersion events that occurred between the Oligocene and the Miocene. We found significant differences in speciation rates between the New World and the remaining Mabuyinae clades only in GeoSSE. The influence of phenotypic traits on diversification rates was not supported by any method. Ancestral state reconstruction suggested that the ancestor of South American mabuyine was likely viviparous.

**Discussion**. Our analyses further corroborated the existence of a transoceanic connection between Africa and South America in the Eocene/Oligocene period (Atlantogea). Following colonization of the isolated South America and subsequent dispersal through the continent by the ancestral mabuyine stock, we detected no difference in macroevolutionary regimes of New World clades. This finding argued against the ecological opportunity model as an explanation for the diversity of living mabuyines.

Corresponding author
Carlos G. Schrago,
carlos.schrago@gmail.com

## INTRODUCTION

The subfamily Mabuyinae comprises 24 genera and 197 species of lizards and belongs to the highly diverse worldwide family, Scincidae (Lepidosauria; Squamata) (*Uetz, Hošek & Hallermann, 2017*). The Mabuyinae, as well as the Scincidae, are distributed on nearly all continents. Approximately one-third of the species in this subfamily are from the Neotropical region, and they are the only representatives of scincids in South America.

The entire diversity of mabuyine species was traditionally assigned to the single genus, *Mabuya*, but a previous analysis proposed four new monophyletic genera with well-defined geographical distributions (*Mausfeld et al., 2002*): (1) *Trachylepis* (previously *Euprepis*), comprising African and Madagascan species with one South American representative (*T. atlantica*); (2) *Eutropis*, containing Asian species; (3) *Chioninia*, from the Cape Verde islands; and (4) *Mabuya*, containing New World species, recently rearranged into 16 new genera by *Hedges & Conn (2012)*. *Hedges & Conn (2012)* treated South American Mabuyinae as a clade of family Mabuyidae, within the superfamily Lygosomoidea. The new Mabuyidae consisted of four new subfamilies: Mabuyinae, Chioniniinae, Dasiinae and Trachylepidinae. Hedges and Conn's arrangement disregarded many genera previously related to Mabuyinae, such as *Eutropis*, *Lankaskincus*, and *Ristella* (*Pyron, Burbrink & Wiens, 2013*), and recent papers have questioned this classification (e.g., *Pyron, Burbrink & Wiens, 2013*; *Pinto-Sanchez et al., 2015*; *Karin et al., 2016*). Recently, alternative classifications were suggested. For instance, *Karin et al. (2016)* placed the Middle-Eastern *Trachylepis* (*T. aurata*, *T. vittatus*, and *T. septemtaeniatus*) into the genus *Heremites* and *Eutropis novemcarinata* into *Toenayar novemcarinata*. Moreover, *Pinto-Sanchez et al. (2015)* assigned species from the genera *Maracaiba* and *Alinea* back to genus *Mabuya* and *Metallinou et al. (2016)* reclassified *Trachylepis ivensii* as *Lubuya ivensii*. In this study, we followed the Reptile Database (*Uetz, Hošek & Hallermann, 2017*) taxonomy as of January 2017, which classified this clade as subfamily Mabuyinae of the family Scincidae.

Therefore, at least two phylogenetically distinct lineages of Mabuyinae are distributed in the New World, namely, *T. atlantica* and the Continental American Mabuyinae (CAM) clade. These lineages are distinguished by both morphological features—presacral vertebrae counts, keeled dorsal scales, coloration, and oviparity (*Greer, 1970*)—and molecular evidence (e.g., *Mausfeld & Vrcibradic, 2002*; *Carranza & Arnold, 2003*; *Whiting et al., 2006*). It is customary to assume that the history of Mabuyinae in South American continent consisted of two independent transoceanic dispersal events from the Old World (*Mausfeld & Vrcibradic, 2002*; *Carranza & Arnold, 2003*; *Whiting et al., 2006*). The spatial distribution of the single representative of the genus *Trachylepis* in an island closer to South America than to Africa, *T. atlantica*, is of particular interest. This issue is so intriguing that it has reached the pages of nontechnical literature (*De Queiroz, 2014*). *T. atlantica* is found in Fernando de Noronha, a small volcanic archipelago in the Atlantic Ocean that lies 375 km

off the northeastern coast of Brazil and that was geologically formed from the Miocene (12.3 Ma) to the upper Pliocene, from 3.3 to 1.7 Ma (*Almeida, 2002*). Although other *Trachylepis* species have spread to several islands near the African continent, the presence of *T. atlantica* in Fernando de Noronha likely represents the farthest dispersal registered for the genus.

The CAM lineage, on the other hand, contains approximately 60 species of Mabuyinae. Previous studies have suggested that the split between this clade and the African Mabuyinae (genus *Trachylepis*) occurred from 28 to 34 Ma (*Hedges & Conn, 2012*; *Karin et al., 2016*). Additionally, the age of the last common ancestor (LCA) of the CAM was dated at 7–9 Ma (*Carranza & Arnold, 2003*; *Pinto-Sanchez et al., 2015*) and 11–14 Ma (*Miralles & Carranza, 2010*; *Hedges & Conn, 2012*; *Karin et al., 2016*). Considering the present-day Atlantic Ocean currents, *Mausfeld et al. (2002)* suggested that *T. atlantica* could have dispersed from the coast of Southwest Africa to South America, but no work to date has comprehensively evaluated this hypothesis. Older ages could be consistent with the supposed faunal transoceanic connection between Africa and South America during the Eocene/Oligocene (e.g., hystricognath rodents, *Loss-Oliveira, Aguiar & Schrago, 2012*; *Voloch et al., 2013*; anthropoid primates, (*Schrago et al., 2012*; *Schrago, Mello & Soares, 2013*); amphisbaenians, *Vidal et al., 2008*; emballonurid bats, *Teeling, 2005*; *Leigh, O'Dea & Vermeij, 2013*; testudinid turtles, *Le et al., 2006*) through a single or a series of islands constituting an Atlantic Ocean Ridge (the Atlantogea paleo province) (*Simpson, 1950*; *Poux et al., 2006*; *De Oliveira, Molina & Marroig, 2009*; *Ezcurra & Agnolin, 2012*; *De Queiroz, 2014*). This transatlantic island corridor, associated with a drop in the sea level in the Oligocene, could explain faunal exchanges in this period (*De Queiroz, 2014*). Regarding *T. atlantica*, there is a lack of estimates of the age of the separation between this species and its African sister lineage.

Chronological information is indispensable for a full understanding of the scenario underlying the current geographic distribution and evolutionary history of extant lineages (*Sanmartín, Van der Mark & Ronquist, 2008*; *Loss-Oliveira, Aguiar & Schrago, 2012*). However, estimates of divergence times within the Mabuyinae have been hampered by the lack of fossils for Mabuyinae, which made previous researchers rely on evolutionary rates borrowed from the literature (e.g., *Carranza & Arnold, 2003*; *Miralles & Carranza, 2010*; *Lima et al., 2013*; *Barker et al., 2015*; *Karin et al., 2016*) or to employ biogeographic events as calibrations (*Hedges & Conn, 2012*; *Pinto-Sanchez et al., 2015*) to derive the timescale of this lineage. The fossil record, however, has been demonstrated to be much more informative than biogeographic events as a calibration tool (*Heads, 2011*). The use of biogeographic events to time-calibrate phylogenies requires the assumption of vicariant scenarios of diversification, which entails that the speciation is synchronous with the breaking apart of landmasses, or that island colonization occurs immediately following geological formation (*Heads, 2011*; *Mello & Schrago, 2012*).

Revealing the origin and diversification of the CAM clade and *T. atlantica* also requires a robust phylogenetic hypothesis. Although it is generally accepted that *T. atlantica* is a member of the genus *Trachylepis* and is therefore excluded from the main diversification of the CAM, its evolutionary affinity remains controversial (*Mausfeld et al., 2002*; *Carranza*
& Arnold, 2003; Whiting et al., 2006). Early proposed phylogenies of Mabuyinae used a maximum of 35 species, a number that significantly underrepresents the diversity of this subfamily (Mausfeld et al., 2002; Carranza & Arnold, 2003; Whiting et al., 2006). Hedges & Conn (2012) studied the CAM clade exclusively and included 40 species, whereas the large-scale Squamata phylogeny of Pyron, Burbrink & Wiens (2013), which contains 4,161 species, sampled 71 mabuyine skink species. Recently, Pinto-Sanchez et al. (2015) used 250 specimens to infer the phylogeny and species diversity of neotropical mabuyinaes, focusing on Colombian populations. Karin et al. (2016) analyzed the higher-order relationships of Mabuyinae using 22 species (24 specimens). Considering that improvements in phylogenetic inference and divergence time estimation can be obtained by increasing taxon sampling (Linder, Hardy & Rutschmann, 2005; Albert et al., 2009; Soares & Schrago, 2012), this matter requires further investigation.

Moreover, the occupation of the South American mainland by the CAM motivates an analysis of differential rates of diversification and rates of phenotypic traits evolution in this clade. It has been broadly reported that the ecological opportunity of a new environment can induce acceleration of macroevolutionary rates (Yoder et al., 2010; Liedtke et al., 2016). This phenomenon was documented for insular vertebrates (Losos & Mahler, 2010; Jönsson et al., 2012); but, as proposed by Simpson (1953), the occupation of a new continent could also trigger evolutionary radiations (Yoder et al., 2010; Pires, Silvestro & Quental, 2015).

In this study, as a means of exploring the continental biogeographic and evolutionary patterns associated with the occupation of South America by mabuyine skinks, we estimated the molecular phylogeny and inferred the divergence times of its members employing fossil information. The inferred time-tree was used to perform, for the first time, a formal statistical analysis of the historical biogeography and macroevolutionary diversification rates of Mabuyinae. To this end, by combining previously published data, we assembled the largest dataset of species and gene sampling composed to date, with the aim of uncovering the evolution of the Mabuyinae.

## MATERIALS AND METHODS

### Data collection, alignment, and taxonomy

We assembled a chimeric supermatrix from previously published sequence data available in GenBank. A total of eight genetic loci from 117 species of Mabuyinae, as well as 103 additional Scincid genera were analyzed, summing 220 taxa, with two genera of the family Xantusiidae used as outgroups. All seven genes available from *Trachylepis atlantica* were used in our analysis: the mitochondrial ribosomal genes *12S rRNA*, *16S rRNA* and the coding gene *cytochrome b* (*cytb*); as well as the nuclear genes alpha enolase (*enol*), oocyte maturation factor (*cmos*), glyceraldehyde-3-phosphate dehydrogenase (*gapdh*), myosin heavy chain (*myh*), and G protein-coupled receptor 149 (*gpr149*). The accession numbers are available in File S1. When several sequences representative of each taxon were available, the longest sequence was selected. In order to simultaneously decrease the number of missing data and to increase the number of representative taxa, we assumed genera and species as monophyletic. Therefore, chimeric supermatrices were used. All protein-coding

**Table 1  General information on the loci used in this study.**

|  | *12S* | *16S* | *enol* | *cytb* | *cmos* | *gapdh* | *myh* | *gpr149* | **Total** |
|---|---|---|---|---|---|---|---|---|---|
| Number of sequenced species | 214 | 194 | 70 | 153 | 146 | 47 | 51 | 73 | 218 |
| Alignment length | 366 | 464 | 192 | 1,033 | 894 | 380 | 133 | 669 | 4,131 |
| Frequency of sites with missing data | 0.12 | 0.20 | 0.74 | 0.38 | 0.40 | 0.81 | 0.81 | 0.72 | 0.47 |
| Frequency of indels | 0.02 | 0.03 | 0.01 | 0.16 | 0.22 | 0.04 | 0.01 | 0.01 | 0.10 |

sequences were visually checked for stop codons and aligned individually in SeaView v. 4.4.3 (*Gouy, Guindon & Gascuel, 2010*) using the MUSCLE v. 3.8.31 (*Edgar, 2004*) algorithm, whereas the ribosomal genes were aligned in MAFFT v. 7 (*Katoh & Standley, 2013*). Gblocks v. 0.91b (*Castresana, 2000*; *Talavera & Castresana, 2007*) was used to exclude poorly aligned bases and divergent regions in the *12S rRNA*, *16S rRNA*, and *enol* genes. Individual genes were then concatenated into a single supermatrix using the R package Phyloch (*Heibl, 2015*). RogueNarok (*Aberer, Krompass & Stamatakis, 2013*) was used to identify taxa without significant phylogenetic information using a ML tree and associated bootstrapped topologies. These datasets were inferred under a rapid bootstrapping algorithm analysis with 200 replicates (*Stamatakis, Hoover & Rougemont, 2008*), followed by a thorough search of the ML tree using the evolutionary model GTRGAMMA, performed in RAxML-HPC (8.1.24) (*Stamatakis, 2014*). Based on the relative bipartition information criterion (RBIC) inferred by RogueNarok, the terminal nodes *Larutia* and *Otosaurus* (RBIC > 1.0), were removed from the analysis. Another dataset, assembled with a more stringent (RBIC > 0.5) criterion, was composed for comparison. Under this criterion, the genus *Lankascincus* and the mabuyine species *Eumecia anchietae* and *Trachylepis acutilabris* were excluded from the analyses. The results, however, were robust for both RBIC values, and we report the results under RBIC > 1.0 hereafter. The final supermatrix consisting of 4,131 base pairs are available in File S2, and detailed information on each locus of the final alignment was listed in Table 1.

## Evolutionary analyses

We investigated 18 candidate partitioning schemes using PartitionFinder heuristic search algorithm with the Bayesian information criterion (BIC) for model selection (*Lanfear et al., 2012*). The partitioning schemes also tested codon positions of protein-coding genes. The partitioning strategy with the best fit consisted of seven partitions (File S3), which were used throughout the analyses. Maximum likelihood (ML) phylogenetic inference was performed in RAxML-HPC (8.1.24) employing the evolutionary model GTRCAT. The GTR substitution model was applied to each partition, as this is the only model supported in RAxML. ML analyses used 200 initial searches for finding the optimal tree topology. Statistical support for clades was assessed using 2,000 standard nonparametric bootstrap replicates (PB).

Inference of node ages was performed with the mcmctree program of the PAML 4.8a package (*Yang, 2007*). For large alignments, the Bayesian inference of node ages via Markov chain Monte Carlo is computationally intensive. To make the analyses feasible, we used an approximate likelihood calculation modified from *Thorne, Kishino & Painter (1998)*

and implemented in mcmctree (*Dos Reis & Yang, 2011*). Priors for the rgene and sigma2 parameters were set as $G(2, 200)$ and $G(1, 10)$, respectively. Markov chains were sampled every 1,000 generations until 50,000 trees were collected. The analysis was performed twice to check for convergence of the chains. Effective sample sizes (ESS) of parameters were calculated in Tracer v. 1.5, and all values were greater than 200.

## Calibration priors

The age of the root, which corresponds to the split between the families Scincidae and Xantusiidae, was calibrated at a minimum value of 70.6 Ma and a maximum age of 209.5 Ma. The minimum was set according to the oldest scincid from the Late Cretaceous of North America (Campanian, 83.5 –70.6 Ma), as previously adopted by *Mulcahy et al. (2012)* (*Rowe et al., 1992*); while the maximum was set according to *Benton, Donoghue & Asher (2015)*, which proposed a maximum age of the ancestral of Squamata. Additional calibration information was gathered from the PaleoBioDB (paleobiodb.org) and entered as minimum ages of the stem nodes of clades. The minimum age of the Scincidae stem node was calibrated at 20.4 Ma, based on the oldest crown scincid *Eumeces antiquus* classified as a member of the subfamily Scincinae (*Holman, 1981*; *Estes, 1983*). The stem node of the clade containing the extant genus *Eumeces* was calibrated at a minimum age of 13.6 Ma based on fossils from the Middle Miocene in North America (*Holman, 1966*; *Voorhies, Holman & Xiang-xu, 1987*; *Joeckel, 1988*). The age of an extinct *Egernia* sp. from the Miocene of Hungary (>5 Ma) was used to calibrate the stem node of the clade containing this extant genus (*Venczel & Hír, 2013*). According to *Böhme (2010)*, the fossil *Tropidophorus bavaricus* belongs to extant genus *Tropidophorus*, and it was used to calibrate the stem node of this clade, setting its minimum age at 13.6 Ma (*Böttcher et al., 2009*). Calibration nodes are shown in the Fig. S4.

## Ancestral area reconstruction

A historical biogeographical reconstruction was performed for the subfamily Mabuyinae and its sister clade (*Lankascincus* and *Ristella*). The R package BioGeoBEARS (*Matzke, 2013*) was used to run likelihood methods: DIVALIKE (a likelihood interpretation of DIVA that allows for the same events as DIVA—*Matzke, 2013*) and DEC (Dispersal-Extinction-Cladogenesis, *Ree & Smith, 2008*). In BioGeoBEARS, we used the likelihood ratio to test whether the null models (DIVALIKE and DEC) fitted the data better than did the more sophisticated models (DIVALIKE + J and DEC + J). The "J" in models represents the addition of the founder-event speciation, thereby allowing dispersal without range expansion (*Matzke, 2014*). The maximum range size, which limits the number of areas defined by tips and nodes, was set to two, based on the current geographic distribution of species. Constraints on dispersal or area availability were not included. To make the biogeographic analysis computationally feasible, islands were not considered independent regions. The rationale for choosing the seven biogeographic areas follows the zoogeographical regions found in the herpetological and biogeographical literature (*Vitt & Caldwell, 2009*; *Lomolino, 2010*; *Morrone, 2014*; *Pyron, 2014*): (1) Neotropical Brazilian Subregion (B), (2) Neotropical Chacoan Subregion (C), (3) West Indies—Caribbean

Islands (W); (4) Oriental Region (O): Southeast Asia + Philippines + Indian Subcontinent (Pakistan to Bangladesh, including Sri Lanka, Nepal, and Bhutan); (5) Afrotropical (A): Sub-Saharan Africa; (6) Madagascar (M): Madagascar and adjacent islands (the Seychelles and the Comoros); and (7) Saharo-Arabian (S): Europe + North Africa + the northern portion of the Arabian Peninsula + Southwest Asia. Using online distributional data from the Reptile Database (Uetz & Hosek, 2015), we classified the tips as belonging to one or more of these areas.

## Rate of species diversification and diversification-phenotype rate correlation

Our dated phylogeny of Mabuyinae was used to infer the dynamics of species diversification using BAMM 2.5 (Bayesian Analysis of Macroevolutionary Mixtures—*Rabosky, 2014*), which simultaneously accounts for variation in evolutionary rates through time and among lineages using transdimensional (reversible-jump) Markov chain Monte Carlo (rjMCMC) (*Rabosky, 2014*). Markov chains were sampled every 1,000th generation until 37,500 trees were collected after a burn-in of 25%. Prior distributions were set according to *setBAMMPriors* function from the BAMMtools R package (*Rabosky et al., 2014*). The frequencies of the species in each genus were considered.

We also tested for trait-dependent diversification. The following traits were collated from the literature: (i) data on the reproductive mode for 60 species of the Mabuyinae—39 species as viviparous, 17 as oviparous, and three as ovoviviparous (*Meiri et al., 2013*; *Pyron & Burbrink, 2013*); (ii) the SVL (snout–vent length) for 46 species (*Meiri, 2010*; *Miralles & Carranza, 2010*; *Das, 2010*; *Hedges & Conn, 2012*; *Meiri et al., 2013*; *Pyron & Burbrink, 2013*); and (iii) the body mass data for 35 species (*Meiri, 2010*; *Hedges & Conn, 2012*). Following previous works, we treated "ovoviviparity" as viviparity (*Pyron & Burbrink, 2013*).

Differences in the rates of speciation ($\lambda$) and extinction ($\mu$) between New World ($NW$) and Old World ($\overline{NW}$) areas and between viviparous ($V$) and non-viviparous ($\bar{V}$) lineages were tested using two approaches. Firstly, an ANOVA, implemented in the R package diversitree (*FitzJohn, 2012*), was used to compare different macroevolutionary regimes. In the $NW/\overline{NW}$ comparison, rates were calculated using the GeoSSE approach (*Goldberg, Lancaster & Ree, 2011*), whereas BiSSE was used to test $V/\bar{V}$ rates, with both tests of binary characters as implemented in diversitree. In this sense, GeoSSE first optimised the parameters under an unconstrained full model, in which ML estimates were obtained for (i) the speciation rate of the New World lineage ($\lambda_{NW}$); (ii) the speciation rate of non-New World lineages ($\lambda_{\overline{NW}}$); (iii) the extinction rate of the New World clade ($\mu_{NW}$); (iv) the extinction rate of non-New World lineages ($\mu_{\overline{NW}}$); (v) the intermediate speciation rate parameter ($\lambda_{NW,\overline{NW}}$); (vi) the dispersal rates from the New World clade ($d_{NW \to \overline{NW}}$); and (vii) the dispersal rates of the sister lineages ($d_{\overline{NW} \to NW}$). Similarly, BiSSE was used to infer (i) the speciation rate of the viviparous lineage ($\lambda_V$); (ii) the speciation rate of non-viviparous lineages ($\lambda_{\bar{V}}$); (iii) the extinction rate of the viviparous clade ($\mu_V$); (iv) the extinction rate of non-viviparous lineages ($\mu_{\bar{V}}$); (v) transition rates from the viviparous clade ($q_{V \to \bar{V}}$); and (vi) transition rates of the sister clade ($q_{\bar{V} \to V}$). Initial parametric values were set according to the *starting.point* function from the diversitree package with an initial

**Table 2  Model comparisons performed in 'diversitree'.** Constraints in the macroevolutionary models subjected to ANOVA for differences in the rates of speciation ($\lambda$) and extinction ($\mu$) between New World (NW) and Old World ($N\overline{W}$) areas (GeoSSE) and between viviparous (V) and non-viviparous ($\overline{V}$) lineages (BiSSE).

| Model | Approach | Constrains | Parameters |
|---|---|---|---|
| Reduced full | GeoSSE | NA | $\lambda_{NW}, \lambda_{\overline{NW}}, \mu_{NW}, \mu_{\overline{NW}}$ |
| | BiSSE | NA | $\lambda_{V}, \lambda_{\overline{V}}, \mu_{V}, \mu_{\overline{V}}$ |
| Equal speciation | GeoSSE | $\lambda_{NW} = \lambda_{\overline{NW}}$ | $\lambda, \mu_{NW}, \mu_{\overline{NW}}$ |
| | BiSSE | $\lambda_{V} = \lambda_{\overline{V}}$ | $\lambda, \mu_{V}, \mu_{\overline{V}}$ |
| Equal extinction | GeoSSE | $\mu_{NW} = \mu_{\overline{NW}}$ | $\lambda_{NW}, \lambda_{\overline{NW}}, \mu$ |
| | BiSSE | $\mu_{V} = \mu_{\overline{V}}$ | $\lambda_{V}, \lambda_{\overline{V}}, \mu$ |
| Equal diversification | GeoSSE | $\lambda_{NW} = \lambda_{\overline{NW}}$ $\mu_{NW} = \mu_{\overline{NW}}$ | $\lambda, \mu$ |
| | BiSSE | $\lambda_{V} = \lambda_{\overline{V}}$ $\mu_{V} = \mu_{\overline{V}}$ | $\lambda, \mu$ |

ratio of 0.5. This preliminary step was required to build the likelihood function (the *make* command). In BiSSE, species with unknown states were coded as 'NA' and assigned the sampling fraction of the species of Mabuyinae used in this work ($\sim$61%), independent of the character state. Subsequently, to perform the ANOVA test of the GeoSSE results, we chose to constrain the intermediate speciation and the dispersal rate parameters to zero ($\lambda_{NW,\overline{NW}} = d_{NW\rightarrow\overline{NW}} = d_{\overline{NW}\rightarrow NW} = 0$), in order to compare speciation and the extinction rates within regions exclusively. Finally, we tested macroevolutionary alternative models against the full models (Table 2).

The second comparison between $NW/\overline{NW}$ and $V/\overline{V}$ macroevolutionary regimes used the marginal posterior distributions of macroevolutionary parameters, inspected using the R package diversitree. These distributions were obtained using the MCMC analyses, with samples taken in diversitree every 1,000th generation until 1,000 samples were collected. A broad exponential prior (mean of 0.5) for all parameters was used, as recommended, while the $\lambda$ and $\mu$ rates were set as the values obtained in the ML full model (*FitzJohn, 2012*). For the $NW/\overline{NW}$ relationship, we also used the marginal posterior distributions obtained from BAMM, applying the *getCladeRates* function of the BAMMtools R package. For all approaches, we calculated the 95% highest posterior density (HPD) interval for the difference between the means of the $\overline{NW}/\overline{NW}$ and $V/\overline{V}$ lineages (*Bolstad, 2007*).

The diversification-phenotype rate correlation was performed in STRAPP (*Rabosky & Huang, 2015*), as implemented in BAMMtools. To run STRAPP, phylogenies were pruned to match the available information for both tree terminals and traits. Diversification analysis results previously obtained using BAMM were pruned to match the available trait information using the function *subtreeBAMM* and were tested against the analyzed traits, using the Mann–Whitney method for binary characters and both Pearson and Spearman methods for continuous traits.

**Table 3  Model comparisons performed in BioGeoBEARS.** Likelihood-ratio tests between null models (DIVALIKE and DEC) and more sophisticated models with the addition of the founder-event speciation (+J) (DIVALIKE + J and DEC + J). (Ln*L*) Likelihood value. (DF) Degrees of freedom for the chi-square test.

| Model | Null model | LnL alt | LnL null | DF | pval |
|---|---|---|---|---|---|
| DEC + J | DEC | −94.34 | −112.8 | 1 | 1.20E−09 |
| DIVALIKE + J | DIVLIKE | −95.65 | −119.2 | 1 | 6.70E−12 |

### Rate of trait evolution and ancestral state reconstruction

We inferred ancestral states of reproductive mode in the BayesTraits software (*Pagel, Meade & Barker, 2004*) and in the R package BiSSE. Both approaches use the maximum likelihood method and allow the use of species with unknown character states. Results were visualized in the R package diversitree. The rates of trait evolution of the two continuous traits studied, namely, SVL and body mass, were inferred in BAMM, using logarithms of both measures. Phylogenies were also pruned to match the available information for both tree terminals and traits. We tested the correlation between SVL and body mass using phylogenetic independent contrasts (PIC; *Felsenstein, 1985*) as implemented in the R package ape. Prior distributions to these subtrees were set according to *setBAMMPriors* function from the BAMMtools R package (*Rabosky, 2014*). Chains were sampled every 1,000 generations until 50,000 trees were collected. The results were summarized and visualized in BAMMtools.

## RESULTS

Mabuyinae was recovered as a monophyletic group (BS = 76). The first split in this subfamily isolated the genus *Dasia* from the remaining Mabuyinae and was inferred to have occurred at 30 Ma, with an HPD interval ranging from 20 to 48 Ma (File S4). The biogeographical model with the highest likelihood was the DEC + J (lnL: −94.34). Therefore, the addition of the J parameter for founder events significantly increased the likelihood of the DEC model ($p = 1.2e^{-9}$, Table 3). Our results supported a model in which the genera *Lankascincus* and *Ristella* split from Mabuyinae (BS = 20) in the Oriental region.

After dispersal from the Oriental region to Africa, the ancestor of the CAM clade split from Saharo-Arabian *Trachylepis* approximately 25 Ma (±16–41 Ma) (BS = 60). Subsequently, the stock that gave rise to New World Mabuyinae dispersed from the Saharo-Arabian region to the Neotropical Brazilian Subregion (Fig. 1). The New World Mabuyinae were recovered as monophyletic (BS = 97), and the age of their LCA was estimated as 19 Ma (12–31 Ma). Our results supported the hypothesis that this lineage reached the Neotropics via the Brazilian Subregion. From this region, the ancestor of *Spondylurus* and *Copeoglossum* dispersed to islands of the West Indies. The continental clade later dispersed along the Brazilian Subregion. Additional dispersal events to the West Indies were inferred to have occurred at least twice: in the ancestor of *Mabuya* and in that of *Alinea* and *Marisora*. *Marisora* was not recovered as monophyletic, with the Central American clade as sister to *Alinea* and the South American clade as sister to *Aspronema*.

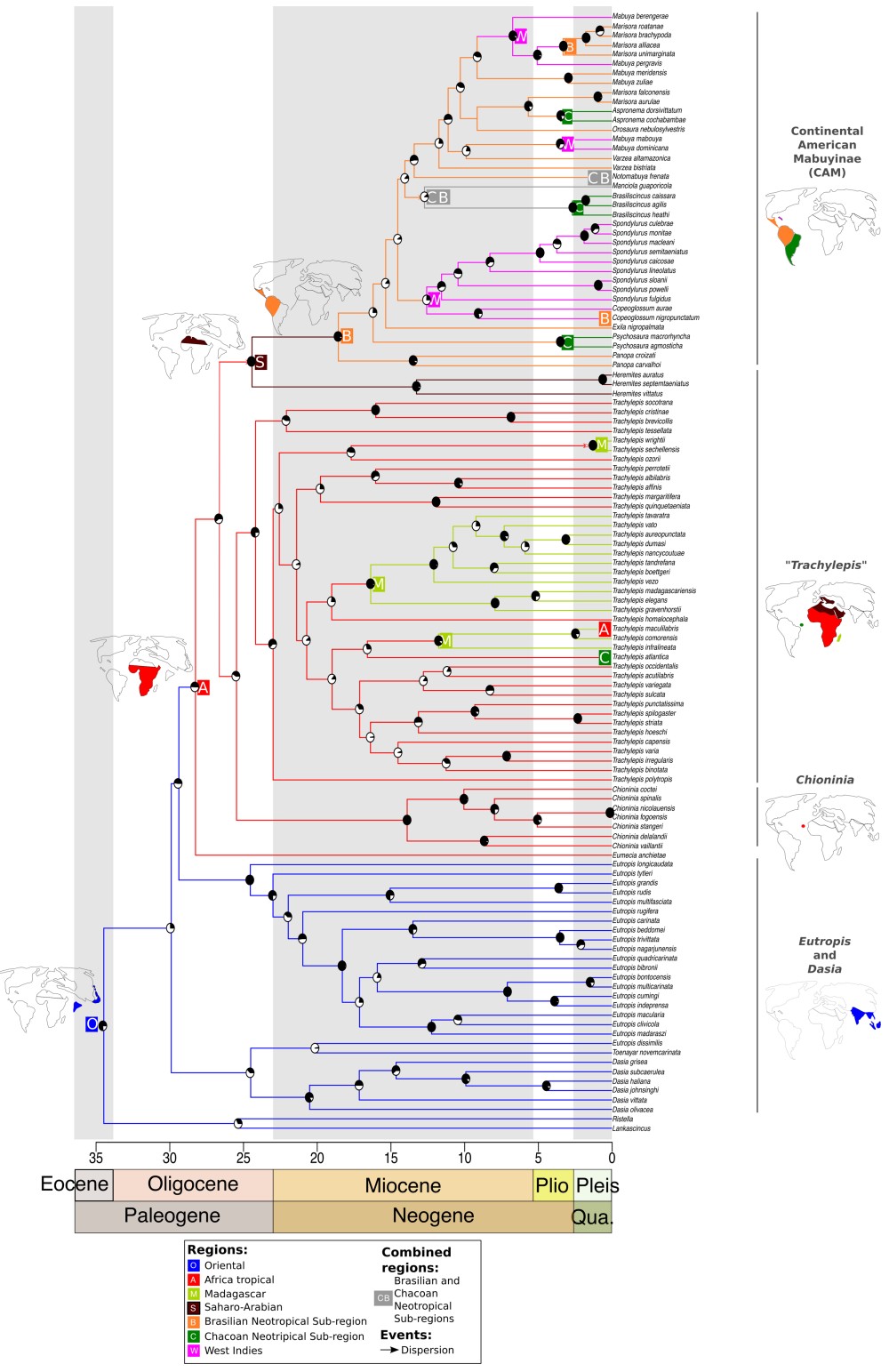

**Figure 1 Time-dated phylogeny of the Mabuyinae with ancestral area reconstruction.** Maps depict putative dispersals and vicariant events that culminated in the occupation of South America by this group. Ancestral area reconstruction was based on the results from DEC + J because this model produced a significantly higher log-likelihood score. The black regions in pie charts represent bootstrap supports.

The origin of *Trachylepis atlantica* consisted of a different history. Firstly, the monophyly of genus *Trachylepis* was not recovered. This genus was split into two lineages: one sister lineage of the CAM clade and another major clade to which *T. atlantica* belongs. Our analysis supported the phylogenetic position of *T. atlantica* as a sister lineage of the clade including the Europa Island *T. infralineata*, the Madagascan *T. comorensis*, and the continental African *T. maculilabris* (BS = 29). We found that the *T. atlantica* ancestor diverged from the remaining *Trachylepis* in the Miocene, approximately 17 Ma (between 10 and 27 Ma) in tropical Africa. The younger diversification of *T. atlantica* indicates that the crossing of this species to Fernando de Noronha occurred more recently than did the occupation of South America by the CAM clade. Our analyses also suggested that once the ancestors of this major *Trachylepis* clade reached Africa, dispersal events from the African continent occurred to Madagascar and nearby islands at least three times, in addition to the dispersal to the Neotropics (*T. atlantica*). Our results showed that a monophyletic Madagascan *Trachylepis* (BS = 94) diverged from its sister group approximately 19 Ma (12–32 Ma) (BS = 23).

The ANOVA was used to test whether the full four-parameter GeoSSE model of the $NW/\overline{NW}$ comparison (model 1) was statistically favored over simpler alternative models (models 2–4). When we constrained speciation rates to be equal (model 2) and to present equal speciation and extinction rates (model 4), the full model was supported over these alternative models ($p = 0.00782$ and $p = 0.02900$, respectively). The comparison with the model that constrained extinction rates (model = 3), however, was not significantly different ($p = 0.15575$).

GeoSSE analysis of the difference between the posterior distributions of speciation rates in $\overline{NW}/NW$ lineages indicated a positive credible interval (CI) (0.005–0.113), suggesting that the speciation rates of South American clades ($NW$) were indeed higher than those of the remaining lineages ($\overline{NW}$). In BAMM, however, we failed to find a significant difference between the posterior distributions of speciation rates (CI: −0.030–0.031). We also found no evidence for diversification shifts during mabuyine diversification. However, the macroevolutionary cohort matrix suggested heterogeneous macroevolutionary rate regimes in the South American Mabuyinae and the remaining clades (Fig. 2A). In both GeoSSE and BAMM analyses, we failed to find differences between the extinction rates (CI$_{geoSSE}$: −0.036–0.074, CI$_{BAMM}$: −0.027–0.027) and between the net diversification rates (CI$_{geoSSE}$: −0.010–0.090, CI$_{BAMM}$: −0.027–0.027) (Fig. 2B).

Concerning the influence of viviparity on diversification rates, no submodels were supported over the unconstrained model ($p > 0.05$). STRAPP analysis failed to find any correlation between diversification rates and mode of reproduction ($p > 0.05$). We also found no differences between the means obtained from the MCMC of the BiSSE (CI: −0.018–0.125, −0.038–0.055, −0.051–0.049, respectively for speciation, extinction, and transition rates).

The ancestor of CAM and their *Trachylepis* sister clade was recovered as viviparous by BayesTraits (P$_{BTML}$ = 0.751; P$_{BTBA}$ = 0.787) and as oviparous by BiSSE (P$_{BiSSE}$ = 1), while the ancestor of CAM alone was recovered as viviparous by all approaches (P$_{BiSSE}$ = 0.953; P$_{BTML}$ = 0.992; P$_{BTBA}$ = 0.983), suggesting that the ancestral species that colonised South

**A**

**B**

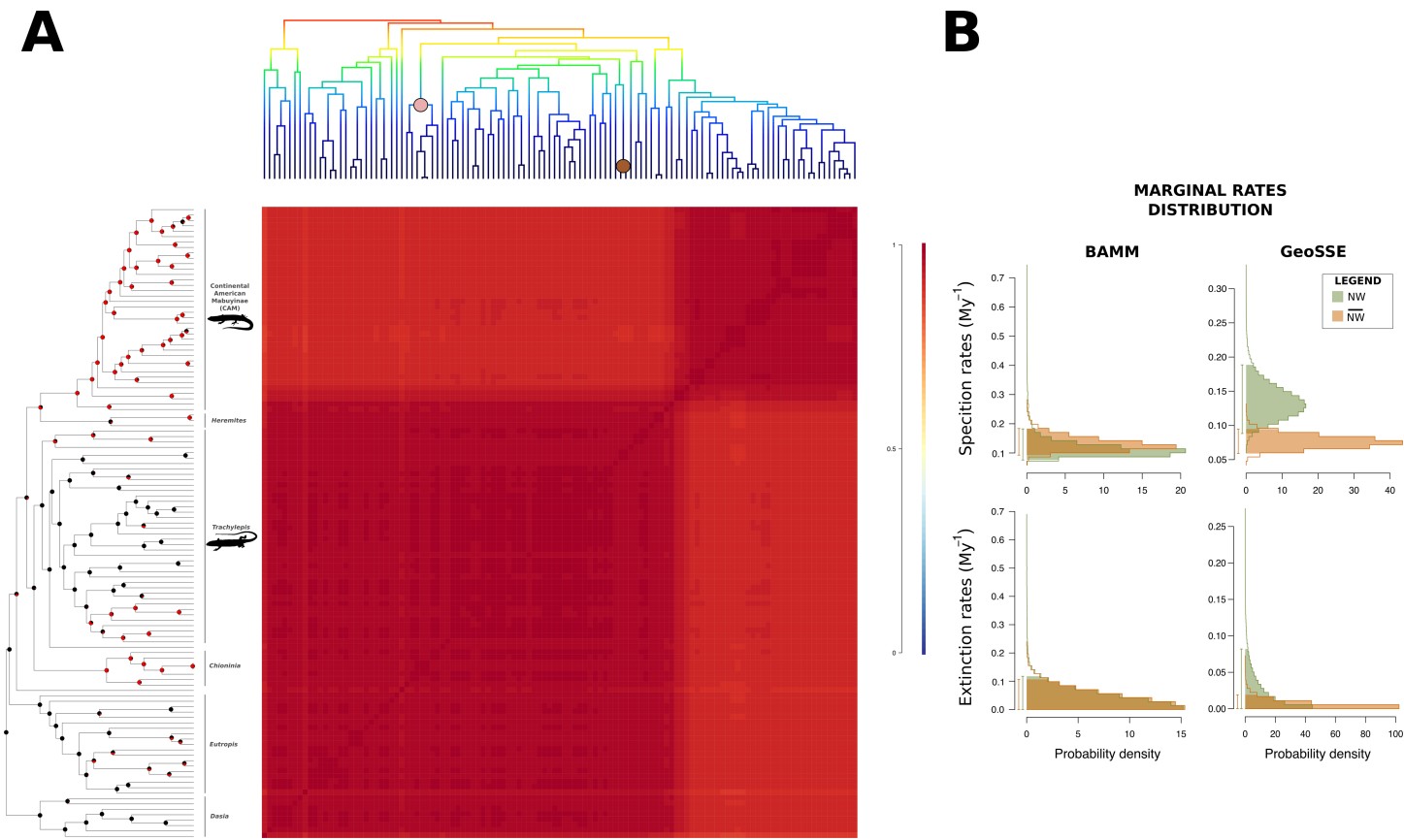

**Figure 2** **Macroevolutionary cohort matrix for the Mabuyinae subfamily.** (A) Illustration of the macroevolutionary rates for the mabuyine lineage obtained using BAMM. For reference, the BAMM tree was plotted at the upper margin of the figure. The pairwise probability that any two species share a common macroevolutionary rate dynamic was indicated by the color of each individual cell. Color scale is indicated at the right. The coral and brown circles in the upper tree represent the shifts found with BAMM in the traits SVL and body mass, respectively. In the left-most tree topology, pie charts at nodes indicate the mode of reproduction inferred by BayesTraits, according to the probability of occurrence of each character. Red represents viviparity, and black represents oviparity. (B) Marginal distributions of macroevolutionary rates inferred in BAMM and GeoSSE were also depicted for New World (green) and non-New World samples (orange).

America was likely viviparous (Fig. 2A). The ancestor of the subfamily Mabuyinae was inferred as oviparous ($P_{BiSSE} = 1$; $P_{BTML} = 0.983$; $P_{BTBA} = 0.980$). Viviparity seems to have emerged several times in the genus *Trachylepis*, as well as in the ancestor of *Chioninia* ($P_{BiSSE} = 0.952$; $P_{BTML} = 0.984$; $P_{BTBA} = 0.971$).

According to PIC results, SVL measurement and body mass were correlated ($p_{LRP} < 0.01$), however, we had found differences in their evolutionary rates. The mean rates of trait evolution of SVL and body mass were smaller in the New World clade than in the $\overline{NW}$ lineages. However, these differences were not significant. In BAMM, we found an increase in the rate of SVL trait evolution in genus *Chioninia*. Only one clade presented an increase in the rate of body mass trait evolution: the Seychelles *Trachylepis* (*T. wrightii* and *T. brevicollis*). STRAPP analysis failed to find a significant correlation between the rate of species diversification and the rate of SVL or body mass trait evolution ($p > 0.05$).

## DISCUSSION

Our analyses based on a fossil-calibrated timetree support the hypothesis that the occupation of South America by the ancestors of the Continental America Mabuyinae (CAM) clade occurred between the Eocene and the Oligocene from the Saharo-Arabian region, corroborating recent estimates (e.g., *Pinto-Sanchez et al., 2015*; *Karin et al., 2016*). This colonization occurred independently of that of *Trachylepis atlantica*, which split from its tropical African sister clade between the Oligocene and the Miocene. Therefore, our results corroborate the hypothesis that the ancestor of *T. atlantica* dispersed from tropical Africa as suggested by *Mausfeld et al. (2002)*. The divergence between *T. atlantica* and its African ancestors was older than the estimated age of the formation of the Fernando de Noronha archipelago (12.3–1.7 Ma; *Almeida, 2002*). Thus, it is unlikely that the ancestors of *T. atlantica* reached South America in a single transoceanic dispersal event. It is more probable to hypothesize a complex dispersal scenario, with a series of dispersals through a transatlantic island corridor leading to Fernando de Noronha, consistent with the Atlantogea model (*De Oliveira, Molina & Marroig, 2009*; *Ezcurra & Agnolin, 2012*). This stepping-stone mode of dispersal could explain the discrepancy between the ages of genetic divergence and the formation of the archipelago because it would dissociate the geological formation of the Fernando de Noronha archipelago from the genetic isolation of the ancestor of *T. atlantica*. This result must be interpreted cautiously because choosing between alternative biogeographical scenarios is influenced by the inferred timescale. The lack of extant and extinct mabuyine species may have impacted divergence time estimates.

The age of the crown node of living CAM species was inferred to be much older than the estimates reported by *Carranza & Arnold (2003)*, 7–9 Ma. The credibility interval of our estimate (12–31 Ma), however, contains the ages inferred by *Miralles & Carranza (2010)*, *Hedges & Conn (2012)* and *Karin et al. (2016)*, which dated this node at approximately 14 Ma. Our results were still inconclusive to establish the precise sequence of events that gave rise to the current geographical distribution of both CAM and *T. atlantica*. Nevertheless, the evolution of these two American lineages of Mabuyinae corroborates a faunal connection between Africa and the Neotropics, and suggests that dispersals may have occurred through island hopping, which is more likely than a single sweepstake event across the Atlantic Ocean. In this sense, it is interesting to mention that a species has been described in the mid-Atlantic Ascension Island by Gray in 1838, which was subsequently assigned to genus *Mabuya*; however, no recent systematic re-evaluation of this specimen is available (*Mausfeld & Vrcibradic, 2002*).

The arrival of Mabuyinae to South America seems to have changed the diversification rate regime of this lineage. The evolutionary rates of both body mass and SVL traits were not significantly altered after the arrival of the ancestors of New World clades, but the increases in rates of both traits were found in island groups, such in Cape Verde's *Chioninia* and Seychelles' *Trachylepis*. The viviparous CAM clade presents peculiar reproductive traits, such as a specialized chorioallantoic placenta that provides fetal nutrition, similarly to eutherian mammals (*Vrcibradic & Rocha, 2011*). This character was suggested to be synapomorphous by *Mausfeld et al. (2002)*. According to our analysis, the CAM ancestor

was already viviparous, in agreement with the large-scale analysis of Squamates (*Pyron & Burbrink, 2013*). However, *Pyron & Burbrink (2013)* suggested an early origin of viviparity in Mabuyinae and multiple transitions to oviparity, including the clade consisting of *Chioninia* and CAM + *Heremites.* On the other hand, possibly because of topological differences, we estimated that the ancestor of Mabuyinae was oviparous and multiple reversions to viviparity occurred later along the evolution of the lineage.

Regarding the biogeographic history of the subfamily Mabuyinae, *Greer (1970)*, based on the primitive characteristics of Asian Mabuyinae, suggested that the ancestors of this subfamily first dispersed from Asia. Other works reasserted this hypothesis, although no formal statistical analyses were conducted (*Honda et al., 1999*; *Honda et al., 2003*; *Karin et al., 2016*). Our results suggest that the early members of Mabuyinae dispersed to Africa and then reached South America and several oceanic islands. The inferred time-dated phylogeny and biogeographic reconstruction similarly suggest an Oriental ancestral distribution, as corroborated by the distribution of the genera *Lankascincus* and *Ristella*, which were recovered as sister clades of Mabuyinae and are geographically distributed in the Oriental region (Sri Lanka and the Indian, respectively). *Karin et al. (2016)* inferred that *Eutropis*, instead of *Dasia*, was the first Mabuyinae offshoot. Nevertheless, as both genera are distributed in the Oriental region, this topological rearrangement would not significantly alter the ancestral area reconstruction.

Finally, this study provides an updated timescale and estimates of macroevolutionary regimes of the diversification of the subfamily Mabuyinae. Our focus was the occupation of the South American continent by the subfamily Mabuyinae through an Oligocene/Eocene transoceanic connection (Atlantogea), which could be responsible for approximately 29% of the South American mammal diversity, suggesting the importance of this event to the extant vertebrate diversity in South America (*Marshall et al., 1982*). CAM and *T. atlantica* are the only representatives of the family Scincidae in South America accounting for 4% of the approximately 1,560 squamate species on this continent (*Uetz, 2000*). Although there are species of other subfamilies of Scincidae in Central America, these species seem to have never crossed the Isthmus of Panama. The converse also seems correct because the only mabuyine genus (*Marisora*) that partially occupied Central America based on our results had arrived via an intermediate dispersion through the West Indies. If this is the case, no representative of Scincidae would be found in present day South American fauna in the absence of this Atlantogea connection. Our results give an overall picture of the timing, biogeography and macroevolutionary dynamics associated with the arrivals of the ancestors of this exceptional case of transoceanic dispersal in two closely related lineages.

## ACKNOWLEDGEMENTS

This study is part of the Doctoral Thesis of AGP from the Genetics Graduation Program at the Federal University of Rio de Janeiro.

### Funding
AGP was financially supported by scholarships from the Brazilian Ministry of Education (CAPES) and from FAPERJ. The funders had no role in study design, data collection and analysis, decision to publish, or preparation of the manuscript.

### Grant Disclosures
The following grant information was disclosed by the authors:
Brazilian Ministry of Education (CAPES).
FAPERJ.

### Competing Interests
The authors declare there are no competing interests.

### Author Contributions

- Anieli Guirro Pereira and Carlos G. Schrago conceived and designed the experiments, performed the experiments, analyzed the data, contributed reagents/materials/analysis tools, wrote the paper, prepared figures and/or tables, reviewed drafts of the paper.

### Data Availability
The raw data has been supplied as a Supplementary File.

### Supplemental Information
Supplemental information for this article can be found online at http://dx.doi.org/10.7717/peerj.3194#supplemental-information.

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
