# Peer review of "Arrival and diversification of mabuyine skinks (Squamata: Scincidae) in the Neotropics based on a fossil-calibrated timetree"

_PeerJ, doi:10.7717/peerj.3194_

## Round 0.1 · original submission · Major Revisions

Although the reviewers tended to be positive, there are a few important issues that need to be address. I'd particularly emphasise the point raised by Reviewer #2 in the sense that you should be more clear regarding the novel aspects of your study with respect to previous studies (particularly those using the similar molecular datasets).

Reviewer 1 ·

Basic reporting

Abstract:
L21. the sister lineage, according to Karin et al. 2016 is Trachylepis + Chioninia, so not exactly distributed across Africa, please clarify this.
L23-24. Be clear which lineage. South American Mabuyinae vs. Trachyelpis atlantica.
L26. You should mention in the methods that your tree is fossil calibrated, which is a main thing that sets it apart from others. Might even be worth mentioning this in the title, because this is some of the more valuable information future researchers will get from your paper.
L30-36. This part of the abstract does not accurately summarize the results and discussion in the paper. For example, you focus the discussion on dispersal via islands, but you don’t talk about that here. Also, you should mention at least what was significant for the things you mention in the abstract methods.

Introduction:
L43. Awkward wording. “…cosmopolitan family, Scincidae”
L48. not really recent at this point.
L50. remove “namely,”
L49-52. Mention more recent taxonomy so that all genera in the subfamily are included. Specifically Karin et al. 2016 - Heremites, Toenayar and Eumecia. Also, Metallinou et al. 2016 - Lubuya. These papers and others: Dasia.
L49-59. Be clear which taxonomy you are using in this paper. i.e. Mabuyinae referring to all of the genera. (see comment in L151-154).
L57-58. be clear Which classification specifically is in question.
L64. Why these 3 citations and not different ones? For example, Hedges and Conn 2012 and/or Pinto-Sanchez et al. 2015. Maybe just include “e.g.,” before the citations to indicate these are just a few of many.
L68. it’s not in South America if it’s on an island. Be clear.
L72. would be interesting to include a divergence time estimate for T. atlantica here.
L77,78,79, etc. Be consistent with date ranges, should be “28–34 Ma”. Also use “en” dashes for ranges instead of hyphens.
L85. sentence fragment. extant _____
L107. Should cite Pinto Sanchez et al. 2015 (for new world timetree) and Karin et al. 2016 (resolved higher-order relationships).
L112-116. mention that you used fossil calibrations for the timetree

Methods:
L126. Be clear that all sequences are from genbank and no sequences were generated de novo.
L130,131. I believe all gene name acronyms should be italicized.
L132-133. be clear, are these chimeric sequences? i.e. different individuals from the same species treated as one individual?
L134,135. please include program version numbers here and in other places as well. example: “MUSCLE v.10.7”
L138. correctly cite the author of phyloch: C. Heibl
L140. How many bootstrap replicates?
L142-143. Combine this sentence with previous, and be clear which program.
L150. Missing a period at end.
L151. Proper database citation: I think: (Uetz & Hosek 2016)
L151-154. move this or something like this paragraph into the introduction around L59.
L155. Delete blank line.
L158. which measure, AIC or BIC?
L159. Did you visually check for stop codons, if so please indicate in prior section in alignment methods
L162. GTRGAMMA or GTRCAT?
L165. Rapid bootstraps or standard bootstraps?
L177. Years in parenthetical citations need to be complete ’92 should be 1992. Fix this throughout this section and other occurrences.
L183-188. Was the calibration always set to the nearest sister node? Please explain in more detail, or in your response. I have not used it, but in mcmctree do you also need to set a maximum age when you specify a minimum? If so, please describe the calibrations
L185-188. Include what ages were set using the Egernia fossil and Tropidophorus fossil.
L210. Cite P. Uetz

Results section: weird extra spacing between paragraphs, fix this.
L285,291,294. etc. use “en” dash for ranges –
L308. IN tropical africa

Discussion:
L364-369. This is not the only possibility. It is also very possible that the closest ancestor of T. atlantica in Africa subsequently went extinct. (see general comment 3.)
L365-366. Citation for this. eg. Geldmacher et al., 2005 or other citations used later in paper
L373. Clarify transoceanic connection. also, ’50 should be 1950
L397,401. Should follow revised taxonomy of Karin et al. 2016. i.e. Heremites aurata and Heremites vittata.
L398. by topology, but with very low support. Karin et al. 2016 showed support for CAM Mabuya with true Trachylepis + Chioninia. This should be mentioned that it conflicts with your topology, though the node in your tree does not show strong support.
L425-426. However, this node is not supported and conflicts with Karin et al. 2016.
L426. Time estimate for what?
L428-429. This seems overly speculative. What Indian route exists? Also, what vicariant event would have been caused by the Himalayas? To my knowledge, there are no species north of the Himalayas in the group.
L431-442. Be careful with wording here. While the hypothesis does not conflict with the time estimate and therefore shows support, there are other possibilities for the phylogeographic scenario. eg. dispersal with source pop. extinction. This section should not make it sound like the data are unequivocal.
L441. should preface sentence with: “If this is the case….”

References:
L467,475, etc… All page ranges should have “en” dashes.
L529. no title?
L545. italicize Trachylepis
L567. italicize atlantica
L571. where is rest of citation?
L600. fix “shifts”
L620. italicize Trachylepis
L635. why 1996 if accessed 2015.
L642. where is the title?


Figures:
Figure 1. the nodes should show some measure of support, i recommend solid circles, open circles, and no circles to represent different levels of support. The map showing phylogeographic implications show arrows that might be misleading. It makes it seem as if the data unequivocally show that Heremites (middle eastern trachylepis) are the source of the main dispersal to south america, however the node is poorly supported in the tree and therefore may not show the true “source” region. especially since other studies show a different sister clade. The taxon names should reflect the most up to date taxonomy. Most importantly, Trachylepis aurata and vittata are now in the genus Heremites. In addition, Eutropis novemcarinata is now in the genus Toenayar (Karin et al. 2016).

Figure 2 Caption. You say “right tree” and i think you mean “left tree”

Additional Figures. I recommend adding another figure or table that displays additional results found.

Supplemental Table SM1. Please include a table caption. What are these accession numbers? They don’t look like genbank accession numbers.
Supplemental Figure SM4. Again, the taxon names should reflect current taxonomy.

Experimental design

no comments

Validity of the findings

Minor issue. See comment 3 and 7 in General comments.

Additional comments

The paper entitled, “Arrival and diversification of mabuyine skinks 1 (Squamata: Scincidae) in the Neotropics,” is valuable addition to the body of literature focused on studying the Mabuyinae. It sets itself apart by being the first study to build a fossil-calibrated timetree within the group and to look in detail about timing and rate of diversification, particularly within the neotropical group.

I recommend several minor as well as more significant changes to the paper, as stated in the line-by-line comments. Furthermore, I recommend the following significant additions:
1.) The fossil calibration methods should be expanded to make it very clear and repeatable. Your logic and reasoning for each fossil time period should be explained.
2.) As this data was compiled from genbank, there are a lot of missing samples, particularly in the nuclear data. Please include in the text, or in a table, the amount of missing data for each locus (i.e. proportion of ingroup taxa with sequence data for that locus, proportion of all taxa with sequence data), and some measure of the amount of informatation in each locus (eg., proportion of variable sites or preferably the proportion of parsimony informative sites).
3.) the authors use wording that makes it seem as though their phylogeographic hypothesis (of dispersal across stepping stone islands) is thoroughly supported by the results. In fact, the results could be due to other explanations, and thus the wording needs to accommodate that.
4.) The abstract should detail the major points of the paper, however the paper covers several other topics not discussed in certain sections of the abstract (see comment below). Please revise the abstract. On this point, be clear in the abstract and introduction that you are testing if the time estimates support the stepping-stone hypothesis across the Atlantic, and what support for that hypothesis would look like in the data.
5.) reiterate in other sections of the paper that this is a fossil calibrated timetree. One could read nearly the entire paper, skipping only the methods, and never know that that was a major goal of the paper. This facet of the research should potentially even be in the title.
6.) the most current taxonomy needs to be used, in particular that of Karin et al. 2016, both in the figures and in the text. (Heremites aurata and Heremites vittata, also Toenayar novemcarinata)
7.) Displaying support values for nodes in phylogenetic trees is important for readers to assess the phylogenetic findings. In addition, discussing ancestral states of nodes that are not supported should be done with caution. As the tree presented here shows some distinct differences to recently published trees that looked at the whole genus regarding higher-order relationships (Karin et al. 2016), the differences may even impact the ancestral-state analysis. I suspect that the nuclear makers (and mtDNA) used in this analysis are not sufficient to resolve the higher nodes. If rerunning the analysis, it may be a wise decision to constrain these nodes to previously published and well-supported trees, especially because levels of support in the analysis in this study are quite low.
8.) Lastly, I believe that the major results of a paper should be shown visually, either in a table or figure. The GeoSSE analysis, and much of the information about what was significant in the diversification rates analyses are only mentioned in the text.

Reviewer 2 ·

Basic reporting

Overall the English writing is ok, but several phrases are too long or have awkward phrasing. See below for suggestions.
Background and the original contributions of the manuscript need to be more clearly stated (see below).

Experimental design

Research question is not clearly well defined. Some methods and approaches used are not properly covered in the contextualization (see more below)

Validity of the findings

Dear editor (and authors),

In the manuscript “Arrival and diversification of mabuyine skinks (Squamata: Scincidae) in the Neotropics (#14440)” authors present an analysis of the phylogeny, biogeography, and diversification rates of South America Mabuyinae skinks. Although the paper address some novel aspects for this group (e.g., estimation of macroevolutionary diversification rates) most of the remaining questions (i.e., the group phylogeny and historical biogeography) have been addressed somewhere else and the dataset used is exclusively based on previously collected sequence data deposited at GenBank. For this reason, I think it is important that authors make it very clear in the Introduction and Abstract what are the real novel aspects of the work and which knowledge gaps they intend to fill. Otherwise the paper just looks like to lack innovation. This issue can also be detected at the Discussion, where phylogenic relations and divergence times are extensively discussed (even though these are not entirely novel, as other studies used the same data for systematics phylogenetics of the group).

Also, if the intention is to present an updated multi-locus phylogeny to be used as a reference on historical biogeography analysis I expected coalescent species tree analysis to be included, not only concatenation methods.
A critical assessment of the taxonomic sampling is not provided…where are the un-sampled species distributed? Without this information it is not possible to evaluate whether the result of higher speciation rates of South American clades (NW) are indeed true or just a sampling artifact (see more below).

Additional comments

Other comments in specific text sections include:

Introduction

Line 48: Instead of ‘recent’ (it is a 2002 study after all) say something about the scope and/or the dataset used in this analysis that split Mabuya.

Line 80-83: Awkward phrasing, please re-phrase.

Line 85: Seems incomplete… extant what? Species?

Line 93: Which events? Not clear, please re-phrase

Line 97: And good estimates of divergence dates also require a robust phylogenetic hypothesis … this phrase is repetitive/circular…re-phrase

Lines 112-119: What new data or approaches were used in the paper? This needs to be clear here. The data was only assembled from previously published dataset or new data was collected? Multilocus? Different analysis? Different calibrations?

This last paragraph of the Introduction needs to be reviewed. As it is now it represents mostly just a list of methods used in the manuscript. Goals and justification/innovation aspects are not clear. If you have a specific hypothesis (e.g., New World Mabuyinae had higher speciation rates) I think this should be stated clearly at the Introduction and be part of the objectives.

Line 124: You assembled data from 117 Mabuyinae species and said in the introduction that there are 190 species in the sub-family. What is the geographic distribution of the unsampled species? This is very important to interpret the result that “speciation rates of South American clades (NW) were indeed higher than those of the remaining lineages” (lines 324 and 325). It is essential to make clear whether the geographic distribution of sampled species is clustered (e.g. non-South American clades) to evaluate whether the result is true or just a sampling artifact.


Line 126: No novel data was collected?

Methods:

If you are not contributing with any new data I think authors should have considered at least a coalescent species tree approach in opposition to a “single concatenated supermatrix” approach

Line 139: “rogue taxa” is not a very common expression, you might consider change for something more usual.

Lines 166-171: Justify the use of PALM for divergence time estimates.

Lines 173-188: There is something wrong with the references cited in this section which does not follow the rest of the document (e.g., (Holman, '66; Voorhies et al., 

185 '87; Joeckel, '88).

Lines 195-198: Wording is strange…re-phrase.

Lines 201-202: I understand the need of computation feasibility (and you have already set the maximum number of areas to 2 to help on that) but what is the biological meaning of not considering islands as independent regions on the biogeographic analysis?

BAMM (Lines 214-221) - Given the criticism that BAMM approach has recently suffered (see Moore et al. 2016. PNAS. Critically evaluating the theory and performance of Bayesian analysis of macroevolutionary mixtures) I expected at least a brief explanation of the method and justification for using it.

Lines 222-229: This is the very first time in the manuscript that the possibility of trait-dependent diversification 
is mentioned (apart from the Abstract)…why do you expect these traits (reproductive mode, body mass, SVL) to influence Mabuyinae diversification? What are your theoretical expectations? Without a contextualization and justification this analysis seems merely exploratory… In other words, authors need to include a better justification for why there were “prompted to test for trait-dependent diversification” and what was the goal of the ancestral states reconstruction for traits.

Are SVL and body mass correlated?

Results:

All bootstrap support values reported are moderate to say the best. It is important that support values are depicted in the manuscript figure (e.g., Figure 1).

Figure 1: include time scale in million years (not only the Geological periods names) and clades support values.

Lines 295-206: Here it is not clear why you claim you results supported the view that the NW Mabuyinae lineage reached the Neotropics via the Chacoan Subregion.

Lines 324-325: See comments above regarding potential taxonomic sampling bias affecting this result. This needs to be addressed in the manuscript.

Line 329: clades instead of clade

Discussion:

Lines 362-363: And when was Fernando de Noronha archipelago formed? References?

Lines 370-374: Too long...re-phrase it. Fix the way references are cited.

Line 405: I would say that the novel aspect of your discussion starts here...

Line 412: How dos the result that ‘the CAM ancestor was already viviparous’ relates with large scale patterns of viviparity evolution in Squamates? (See: Pyron & Burbrink, 2013).

Line 413-414: Why changes in reproductive mode seem to be correlated
with dispersals or island occupation? This is not stated at the Results.

Line 432-433: It would be desirable to point out mechanisms for the supposed Oligocene/Eocene transoceanic connection (E/O). The way this event is explored in the manuscript is kind of speculative and based only on patterns. Also, is the term (E/O transoceanic connection) coined by other authors or proposed in this manuscript? It is not clear…

Line 434: ‘possible’ event

Much of the discussion is devoted to discuss the phylogeny and divergence times, however, these are not entirely novel, as other studies used the same data for systematics phylogenetics of the group.

References:

Line 529: Manuscript title and complete reference?

Uetz, 1996: Use more recent numbers... The Reptile Database is constantly updated, no reason to use numbers from 1996 or even 2015.

---

## Round 0.2 · Minor Revisions

I believe that you properly addressed all of the issues raised by Reviewer 2. However, there are a few issues that were identified by a second round of review by Reviewer 1 that are included below. However, as you'll notice, these differences are fairly straightforward and it shouldn't take long to incorporate them into your manuscript.

Reviewer 1 ·

Basic reporting

Some grammatical issues still present, but overall much better than in first submission. Otherwise very good.

Experimental design

The adjustments to the introduction explaining the research scope in more detail were sufficient and well-written.

Validity of the findings

Revisions were good.

Additional comments

Overall, the revisions made were appropriate and substantially improved this manuscript. I now can confidently recommend this paper for publication, pending the following minor changes. The only more major change is to the paragraph in L416-427, which should be adjusted to be more clear.

L55. This has been updated since this revision in reptile database. Now 24 genera and 197 species.
L56. Change to “…highly diverse worldwide family, Scincidae.
L59. Not exactly true that they are the only skinks in the neotropics. I think you mean in South America.
L107. Delete comma after citation.
L142. Change to “…analysis of diversification rates of phenotypic traits…”
L147. Delete “theoretically”
L152. Change “performed” to “perform”
L152-3. Change to “…a formal statistical analysis of the historical biogeography and macroevolutionary diversification rates of Mabuyinae.”
L154. Change “data-mining” to “combining”.
L161-163. Incorporate L172 “chimeric supermatrix” into this first sentence, as well as the fact that the sequences are from previous datasets. Also combine with some info from next 2 sentences. e.g., “We assembled a chimeric supermatrix from previously published sequence data on genbank for eight genetic loci from 117 species of Mabuyinae, as well as 102 additional Scincid genera. This dataset totaled 219 taxa, and two genera of the family Xantusiidae were used as outgroups.”
L163. Table says 218 species, which is it?
L171-172. Please expand the justification of the use of chimeric sequences.
L174, 175, 182. Be consistent with how you write version numbers. Follow L174.
L183-184. Based on what analysis were they removed?
L227. Change to “Calibration”. Thanks for including this in the figure!
L262. Indent. Also, change to “We also tested for trait….”
L311. Change to “reproductive mode”
L354. Change to “dispersal”
L358. In Figure 2. (NW) ̅ (Old World, bar on top) actually show a higher speciation rate. Is the legend wrong?
L402. Change to “corroborate”
L416. 7–9 should have no spaces.
L416-427. This paragraph needs to be revised to be more clear. Unclear what you mean by incompatible. (elaborate?). What are you trying to say based on the fossil in Ascension? the sequence of ideas doesn’t flow very well here.
L429. Change to “Evolutionary”
L433. Delete “the”
L437. Delete “analyses”
L440. “reversions” is not the right word because the ancestor was oviparous in your analysis. Change to “transitions” or “independent derivations”.
L442. This paragraph is not really about higher systematics, but about biogeography of the whole subfamily. Change the topic sentence.
L445, 454, 467. Use present tense “suggest”, “provide”, “present”
L469. Depends what you call a clade (Scincidae? Mammalia? are both clades). And what you call transoceanic (going to Madagascar?/australia). I recommend ending it with a more general statement saying it is an exceptional case of transoceanic dispersal in closely related lineages.
L597. Title is incorrect in this citation.
L648. change hyphen to 'en' dash in page number. Please double check all refs.
L656. update "accessed" date
Table 1. Confirm is it 218 loci? (see L163)
Figure 2B. Include in caption what is the NW and Old World labels/colors.

---

## Round 0.3 · accepted · Accept

I am happy with the final modifications to the manuscript.